# The Peculiarity of Infection and Immunity Correlated with Guillain-Barré Syndrome in the HIV-Infected Population

**DOI:** 10.3390/jcm12030907

**Published:** 2023-01-23

**Authors:** Yanli Wang, Jun Yang, Ying Wen

**Affiliations:** 1Department of Infectious Diseases, The First Affiliated Hospital of China Medical University, Shenyang 110001, China; 2Neurology Department, The First Affiliated Hospital of China Medical University, Shenyang 110001, China

**Keywords:** HIV, infection, immunity, GBS

## Abstract

Guillain-Barré syndrome (GBS) can occur at all stages of human immunodeficiency virus (HIV) infection. HIV, cytomegalovirus (CMV), and varicella zoster virus (VZV) are the main infectious agents in HIV-positive GBS cases. These cases include acute and chronic HIV infection, immune reconstitution inflammatory syndrome (IRIS) shortly after anti-retroviral therapy (ART), those with ART interruption, or those with cerebrospinal fluids (CSF) HIV escape. The mechanisms are involved in both humoral and cellular immunities. Demyelinating and axonal neuropathies are the main pathological mechanisms in GBS. Presentation and prognosis are identical to those in patients without HIV infection. Typical or atypical clinical manifestations, CSF analysis, electrophysiological and pathological examination, and antiganglioside antibody detection can help diagnose GBS and classify its various subtypes. Intravenous immunoglobulin and plasma exchange have been used to treat GBS in HIV-positive patients with a necessary ART, while ganciclovir or foscarnet sodium should be used to treat ongoing CMV- or VZV-associated GBS. Steroids may be beneficial for patients with IRIS-related GBS. We reviewed HIV-positive cases with GBS published since 2000 and summarized their features to highlight the necessity of HIV testing among patients with GBS. Moreover, the establishment of a multidisciplinary team will guarantee diagnostic and therapeutic advantages.

## 1. Introduction

Guillain-Barré syndrome (GBS) is a common cause of acute flaccid paralysis, and is generally regarded as a post-infection, immune-mediated disease. The preceding infections usually resolve before the occurrence of limb weakness. Infectious agents, including Campylobacter jejuni, cytomegalovirus (CMV), Epstein-Barr virus (EBV), Mycoplasma pneumoniae, Haemophilus influenzae, varicella zoster virus (VZV), influenza A virus, human immunodeficiency virus (HIV), Zika virus, hepatitis E virus, and the severe acute respiratory syndrome coronavirus 2, have been linked to GBS [1]. To date, the prevalence of GBS in the HIV-infected population remains scarce. GBS can present at all stages of HIV infection, and this kind of immune dysfunction could be HIV-induced, anti-retroviral therapy (ART)-induced, or correlated with an opportunistic infection. We reviewed reports of cases of HIV infection with GBS published since 2000 and summarized their features.

## 2. Method

PubMed and Web of Science databases were searched for articles published after 2000, using the following terms: (“Guillain-Barré syndrome” or “GBS”, “Inflammatory Demyelinating Polyneuropathy” or “paralysis”, “Acute motor axonal neuropathy”, “Miller-Fisher syndrome” or “ataxia and nystagmus”) and (“Human Immunodeficiency Virus” or “HIV” or “AIDS” or “immune reconstitution”). The language of articles was limited to English. Studies with reported cases were included, while studies with incomplete data or without reported cases were excluded (Figure 1). Direct drug toxicity, such as stavudine-associated GBS-like manifestation, was excluded [2,3,4,5]. Finally, a total of 41 cases from 32 studies were included. The date of completion of the data collection was 1 November 2022.

### 2.1. The Peculiarity of Infectious Agents and Pathogenesis

HIV infection is recognized as a significant antecedent to GBS infection [6]. In most cases, positive plasma HIV RNA at the time of GBS presentation suggests a para-infectious disease mechanism. CMV and VZV infections are commonly associated with opportunistic infection-correlated GBS [7,8]. Furthermore, pegylated interferon alpha 2b-associated GBS [9], and influenza virus A/H1N1 vaccination-induced GBS [10] have been reported in the HIV-positive population. It is well known that HIV infection, drugs used in HIV-positive cases, as well as opportunistic infections, can either cause polyneuropathy, infective peripheral neuritis, or GBS. For a definitive diagnosis, cases must meet the diagnostic criteria for GBS, such as the Brighton criteria laid out by the Brighton Collaboration or the criteria of National Institute of Neurological Disorder and stroke (NINDS) [11,12]. The Brighton criteria demonstrated clinical case definitions of classic GBS and Miller-Fisher syndrome (MFS). The NINDS criteria list features required for diagnosis, features that strongly support diagnosis and features that cast doubt on diagnosis. GBS is an autoimmune disease of the nerve conduction failure, whose pathogenesis is mainly based on an autoantibody-mediated effector pathway, while partial involvement in a cell-mediated pathway. The mechanism of Campylobacter jejuni-related acute motor axonal neuropathy (AMAN) is triggered by molecular mimicry due to structural similarities between peripheral nerves and C jejuni strains, followed by activated complement deposits [1]. The main target injury component is the nerve axon in AMAN. The demyelination and lymphocyte infiltration are often observed in patients with acute inflammatory demyelinating polyradiculoneuropathy (AIDP) [13]. The macrophage-associated demyelinating lesions in AIDP could be located at both the internodes and at the nodes of Ranvier [14]. Cytokines drive cellular immunity and inflammation in GBS [15]. Similarly, there is a possible link between CMV glycoconjugates, peptides, myelin proteins, and Schwann cell epitopes [16]. Immune checkpoint inhibitor-induced GBS likely has a T-cell-mediated pathogenesis [1]. CMV- or EBV-induced GBS usually trigger demyelinating neuropathy [1]. The pathogenesis of GBS in HIV infection remains largely unknown. The potential mechanisms are as follows: cell-mediated immunity, including macrophage-related demyelination and perivascular T lymphocytic infiltration [17,18]; antibody-mediated damage on nerve axolemma driven by molecular mimicry [8,18]; opportunistic infections such as CMV and VZV infection [8,16]; the indirect neurotoxicity through HIV-proteins [19,20,21]. Although genetic susceptibility favors the development of GBS [22], no specific genetic risk loci have been clearly defined [22].

### 2.2. The Peculiarity of Immunological Spectrum and Clinical Relevance

GBS occurs mostly during seroconversion, especially in acute retroviral syndrome (ARS) or early asymptomatic stages (Table 1). Among the HIV-positive cases, C. jejuni infection is an uncommon cause of GBS, which was only verified in one case [23]. HIV itself could be an antecedent infection before disease onset and is identical to the causative agents of gastrointestinal and respiratory infection. ARS is associated with high viral levels, CD_4_^+^ T cell reduction, and immune activation [24]. The common manifestations of ARS are febrility, slight gastrointestinal or respiratory symptoms, and maculopapular eruptions, usually reported 4–30 days before the GBS onset in five cases [7,25,26,27,28]. Although neurological dysfunction during primary HIV infection is related to high HIV RNA load in the cerebrospinal fluids (CSF) [29], there are insufficient data on CSF HIV RNA in HIV-positive GBS cases [28,30,31,32]. GBS in the HIV-positive population also occurs in patients with chronic HIV infection (Table 2). Among cases with CD_4_^+^ T cell counts >200 cells/μL, GBS may be associated with high HIV-load-related immune dysfunction. Among cases with CD_4_^+^ T cell counts of <50 cells/μL, GBS is possibly correlated with opportunistic infections, such as CMV infection, which may be verified by positive CMV-DNA in the CSF in one case, as detected by a polymerase chain reaction assay [16]. CMV infection is usually a primary infection among pediatric GBS cases [33], while CMV reactivation usually occurs in adults. A recent herpes zoster attack (reactivation), especially within 1–42 days before GBS onset [34], has also been reported in one HIV-positive case [8]. In addition, GBS may also occur shortly after ART as immune reconstitution inflammatory syndrome (IRIS). The IRIS criteria are: (1) presentation of new (or improved then deteriorated) signs and symptoms, (2) an increase in CD_4_^+^T cell counts and (3) a decrease in HIV viral load after ART initiation. The pathogenesis of IRIS involves cell-mediated immunity, such as CD_4_^+^ Th1^+^ effector cells, CD_8_^+^ T-cells, and an imbalance of Th17 and Treg cells [35]. The seven cases with IRIS-GBS commonly presents at 1–2 months post- ART [19,23,36,37,38,39,40]. The cellular immune response during GBS is increased by the release of cytokines, such as IL-17 from activated Th17 cells [41]. GBS can also occur when ART is discontinued in four cases [17,31,32,42], and GBS episodes coinciding with rebound HIV often present within three months of ART interruption. Another possible mechanism of GBS is CSF HIV escape, which as reported in one case [31]. HIV enters the brain during the early stages of an acute infection, and HIV RNA is frequently found in CSF- of HIV-infected patients who do not receive ART. However, CSF HIV escape requires further attention. CSF HIV escape occurs via CNS reservoirs or cell trafficking through the CNS, which is characterized by a higher HIVRNA load in the CSF than in the plasma [43]. However, there is currently insufficient evidence of CSF HIV escape in cases with GBS.

### 2.3. The Peculiarity of Presentation and Prognosis

GBS usually presents as a monophasic course with symptoms peaking within four weeks of onset, followed by a recovery course that persists for months or years, and was identical between patients with and without HIV infection. Furthermore, recurrent GBS episodes [17,31] and chronic inflammatory demyelinating polyradiculoneuropathy [7,48,56], have been reported among the HIV-positive population. Most previously reported HIV-positive GBS cases (31 cases) were found in men, all younger than 60 years. The annual incidence rate of GBS increases with age and is slightly higher in males than in females among general population [18]; two patients had a history of GBS [19,47]. Most HIV-positive patients with GBS (36 cases) survived, although some patients had sequelae. GBS dysfunctions can be motor, sensory, or autonomic, presenting heterogeneous clinical manifestations, including several subtypes. Nerve conduction studies can distinguish between various subtypes. AIDP (18 cases verified by nerve conduction studies) is the most common GBS type, followed by AMAN [8,49,53,54], MFS [10,39,50], and acute motor and sensory axonal neuropathy (AMSAN) [27,40,52]. Autonomic disturbance-related cardiac arrest and respiratory failure requiring mechanical ventilation (seven cases) are the most common causes of death (four cases) [7,17,26,37,44,45,48,52]. Albuminocytologic dissociation (ACD) (33 cases) is a CSF feature typical of GBS where CSF protein levels are elevated due to myelin sheath or axon injury, resulting in increased protein synthesis and release into the CSF, while the CSF total white cell count <50 cells/µL. However, ACD is absent in four HIV-positive patients (CSF total white cell count ≥50 cells/µL) [28,31,49,50], most likely related to positive CSF HIV RNA. Spinal images may show hyperintense and hypertrophic nerves, especially at the nerve root, on gadolinium-enhanced magnetic resonance imaging (MRI) [57]. Typical MRI manifestations are observed in two HIV-positive GBS cases [47,54]. MRI is not a routine diagnostic tool for GBS, whose features are nonspecific, but can be helpful for a differential diagnosis [58]. The antiganglioside antibodies (AGA) are nonspecific for GBS diagnosis, and AGA profiles have never been not determined in HIV-infected patients with GBS [59,60]. AGA test was only applied to eleven HIV-positive patients, including seven cases with positive results [8,10,31,32,38,39,50]. The IgG class switch of AGA could be T-cell-independent or involve T cells [8]. The anti-GM1 and GD1a antibodies are involved in *C. jejuni*-related GBS, particularly AMAN [8,37]; anti-GM2 IgM is implicated in GBS with CMV infections [16], and MFS is closely associated with anti-GQ1b IgG antibodies [39,50]. The specific anti-nerve autoantibody biomarkers in AIDP have not been determined. HIV infection results in the abnormal immunoregulation of CD_4_^+^ T cells, which has an important impact on the selection of self-reactive antibody repertoires and may contribute to pathological autoimmunity [61]. The titer of AGA usually peaks in the progressive phase and gradually decreases thereafter [18]. Although the diagnostic value of AGA is limited [18], a positive test result can be helpful especially when the diagnosis of GBS is doubtful.

### 2.4. The Peculiarity of Treatment Approaches

Firstly, plasmapheresis (plasma exchange) and intravenous immunoglobulin (IVIG) were applied to treat GBS in patients with HIV, which should be started as soon as possible. Spontaneous recovery was reported in a case of AMAN [49]. Generally, patients with pure MFS tend to present with a mild disease course, and usually spontaneously recover within six months. Steroids alone are not recommended but could be advantageous when used in combination with IVIG [1,55]. Secondly, among cases of IRIS-related GBS, which is associated with cell-mediated immune mechanisms [35], steroid initiation was applied to two cases [19,36]. Thirdly, ART should be considered in HIV-positive GBS cases. ART is beneficial, especially in cases of primary HIV infection characterized by an extremely high HIV RNA load. ART can result in immune reconstitution with the upregulation of CD_4_^+^ T-regulatory cell counts, which is beneficial for the remission of autoimmune phenomena. An ART regimen with good CNS penetration will favor a rapid reduction in CSF HIV RNA levels. Fourthly, ganciclovir or foscarnet sodium should be considered in cases of CMV- or VZV-associated GBS. Finally, during the progressive phase, patients with the possibility of developing respiratory failure and requiring ventilation should be promptly transferred to an intensive care unit.

## 3. Conclusions

GBS is uncommon in the HIV-positive population. It could be the first sign of HIV infection [55], which highlights the necessity of HIV testing among patients with GBS. In addition, GBS should be considered when acute limb weakness occurs in individuals with HIV infection. Screening opportunistic infections and assessing the immunological status among HIV-positive GBS cases will guide us in tailoring individual treatments. Importantly, a multidisciplinary team, including HIV specialists, intensive care unit physicians, and neurologists, would improve diagnostic and therapeutic advantages. The performance of prognostic models—such as the modified Erasmus GBS Outcome Score for predicting the inability to walk unaided at 6 months [62] and the Erasmus GBS Respiratory Insufficiency Score for identifying high risk for mechanical ventilation [63]—should be further validated among HIV-positive individuals.

## 4. Future Directions

In order to pursue a favorable outcome, potential medicines that could target autoantibodies and membrane attack complexes, as well as promoting neural regeneration, are in development [64]. Morphological studies using sonography showed the advantage of detecting a gradual improvement in GBS by evaluating the change in the nerve size [65]. Future research directions should address unsolved issues, such as the immunopathological mechanism, specific biomarkers, valuable prognosis assessment models and optimal treatment approaches.

## Figures and Tables

**Figure 1 jcm-12-00907-f001:**
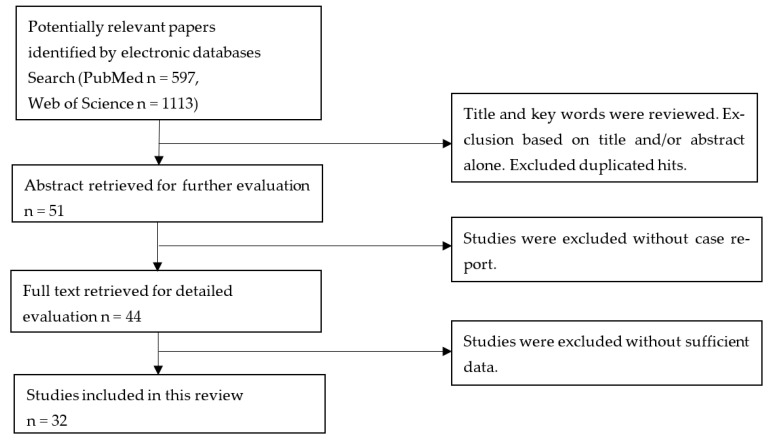
Flowchart of the study selection process.

**Table 1 jcm-12-00907-t001:** The primary HIV-infected cases with GBS.

Numbers of Cases	Reference Number/Year of Publication	Age (yr)	Gender	Clinical Presentations	GBS History	Brain/Spinal Cord MRI or CT	CSF Cells/Protein (Cells/mm^3^)/(mg/L)	AGA	HIV Stage	CD_4_^+^T Cell Counts/HIV RNA Plasma/HIV RNA CSF	Correlation with ART Interruption	IRIS/Post ART	Treatment	Outcome D/S/I
1	[44]/2003	56	F	AIDP * (RD)	No	NM	20/246.8	NM	acute	NM/NM/NM	No	No	Plasmapheresis/MV	S
1	[45]/2003	21	M	AIDP * (RD)	No	NM	30/980	NM	acute	NM/NM/NM	No	No	Plasmapheresis/MV	S
1	[17]/2006	38	M	AIDP * (RD)	No	NM	8/1940	NM	Acute Chronic	502/NM/NM 570/2600/NM 677/380,000/NM	No Yes/15 d Yes/30 d	No No No	IVIG/MV/ART ART ART/IVIG	S
1	[23]/2007	26	M	AIDP * (Campylobacter jejuni)	No	NM/Normal	3/29.8	negative	acute	125–150/50,000-0/NM	No	Yes/2 m	IVIG/ART	S
1	[25]/2008	30	M	AIDP*	No	Normal/Normal	26/720	NM	Acute (ARS)	408/>100,000/NM	No	No	IVIG/ART	S
1	[26]/2011	56	M	GBS	No	Normal/Normal	3/291	NM	Acute (ARS)	710/4400/NM	No	No	IVIG	D (Cardiac arrest)
1	[46]/2014	48	M	AIDP *	No	NM	25/740	NM	acute	400/NM/NM	No	No	IVIG	S
1	[7]/2015	33	M	AIDP * (RD) (mycoplasma)	No	Normal/Normal	35/109	NM	Acute (ARS)	526/2,095,380/NM	No	No	ART/IVIG/levofloxacin/Plasmapheresis/MV	S
1	[27]/2017	57	F	AMSAN *	No	Normal/NM	39/1690	negative	Acute (ARS)	730/195,328/NM	No	No	IVIG/ART	S
1	[36]/2017	38	M	AIDP *	No	NM	ACD	NM	acute	90/57,000/NM 175/590/NM	No	No Yes/15 d worsening	IVIG/ART /GC	S
1	[47]/2019	23	M	AIDP *	Yes (Lyme disease)	Normal/Abnormal	0/320	NM	acute	574/>500,000/NM	No	No	IVIG/ART	S
1	[28]/2021	60	F	AIDP *	No	Normal/Normal	67/1148	NM	Acute (ARS)	334/>10,000,000/1,230,000	No	No	IVIG/ART	S

AGA, antiganglioside antibody; CSF, cerebrospinal fluids; ART, anti-retroviral therapy; F, female; M, male; AIDP, acute inflammatory demyelinating polyradiculoneuropathy; AMSAN, acute motor and sensory axonal neuropathy; *, verified by nerve conduction studies; NM, not mentioned; D, died; S, survived; I, ineffective; IVIG, intravenous immunoglobulins; GC, glucocorticoid; MV, mechanical ventilator; MRI, magnetic resonance imaging; CT, computed tomography; Abnormal, abnormal enhancement of the cauda equina roots; RD, respiratory distress; ACD, albumincytological dissociation; ARS, acute retroviral syndrome.

**Table 2 jcm-12-00907-t002:** The chronic HIV-infected cases with GBS.

Numbers of Cases	Reference Number/Year of Publication	Age (yr)	Gender	Clinical Presentations	GBS History	Brain/Spinal Cord MRI or CT	CSF Cells/Protein (cells/mm^3^)/(mg/L)	AGA	HIV Stage	CD4^+^T Cell Counts/HIV RNA Plasma/HIV RNA CSF	Correlation with ART Interruption	IRIS/Post ART	Treatment	Outcome D/S/I
1	[16]/2001	37	M	GBS *(CMV)	No	Normal/NM	1/3170	NM	Chronic	66/2,400,000/NM	No	No	ART/Ganciclovir	S
1	[30]/2002	35	M	AIDP *	No	Normal/NM	14/740	NM	Chronic	149/745,400/1,41600	No	No	ART	S
1	[19]/2002	56	M	AIDP *	Yes (Diarrhea, HIV?)	NM	0/1393	NM	Chronic	86–510/217,075-<50/NM	No	Yes/1 m	ART/IVIG/GC	S
10	[48]/2003	22–55	M6/F4	GBS/CIDP3?	No	NM	0–17/elevated(9)	NM	Acute(1)/chronic(9)	>200(6),<200(4)/NM/NM	No	No	Plasmapheresis(5/5)/IVIG(3/4)/MV1	D1(cardiac arrest)/S9
1	[37]/2003	58	M	AIDP *	No	Normal/Normal	4/580	NM	Chronic	31–602/867,736–2685/NM	No	Yes/26 d	Plasmapheresis/stopping ART/MV	D (pneumonia)
1	[31]/2005	35	M	AIDP * (HIV escape)	No	NM	85/NM	NM positive	Chronic	914/2999/567 NM/84/307	No Yes/2 m	No No	ART ART/IVIG	S
1	[49]/2007	46	M	AMAN *	No	NM	54/282	negative	Chronic	150/NM/NM	No	No	No	S
1	[50]/2007	56	M	MFS * (Cryptococcus neoformans)	No	Abnormal/NM	84/105	positive	Chronic	24/46,000/NM	No	No	IVIG/ART	S
1	[8]/2011	33	M	AMAN * (Herpes zoster)	No	Normal/Normal	6/170	positive	Chronic	334/7900/NM	No	No	IVIG/ART	S
1	[51]/2011	45	F	AIDP *	No	Normal/Normal	39/240	NM	Chronic?	334/394,000/NM	No	No	IVIG	S
1	[10]/2012	58	M	MFS * (A/H1N1vaccination)	No	Abnormal/NM	3/590	positive	Chronic	310/<40/NM	No	No	ART/IVIG	S
1	[38]/2014	36	M	AIDP *	No	Normal/Normal	0/970	positive	Chronic	545–517/212,000-116/NM	No	Yes/2 m	IVIG/ART	S
1	[52]/2014	32	M	AMSAN *(RD)(enteritis)	No	Normal/NM	ACD	NM	Chronic	71/NM/NM	No	No	IVIG/MV/Plasmapheresis/ART	D (sepsis/MODS)
1	[53]/2014	11	F	AMAN * (dysentery)	No	NM/Normal	Normal	NM	Chronic	542/NM/NM	No	No	IVIG	S
1	[42]/2015	21	M	AIDP *	No	NM	19/5268	NM	Chronic	613/209,720/NM	Yes/3 m	No	IVIG	S
1	[54]/2015	22	F	AMAN * (*Escherichia coli*)	No	NM/Abnormal	0/950	NM	Chronic	5/NM/NM	No	No	IVIG	S
1	[39]/2017	46	M	MFS	No	Normal/NM	9/737	positive	Chronic	750–1053/<40/NM	No	Yes?/7 m (change ART)	ART/IVIG	S
1	[32]/2020	NM	M	AIDP	No	Normal/Normal	33/920	positive	Chronic	700/31,207/939	Yes/45 days	No	IVIG/ART	S
1	[55]/2021	52	M	AIDP *	No	NM/Normal	0/680	NM	Chronic	219/97,800/NM	No	No	IVIG/ART/GC	S
1	[40]/2022	34	M	AMSAN * + BBE	No	Abnormal/Normal	2/1090	negative	Chronic	160–372/<50/<50	No	Yes/3 m	IVIG	I

AGA, antiganglioside antibody; CSF, cerebrospinal fluids; ART, anti-retroviral therapy; F, female; M, male; AIDP, acute inflammatory demyelinating polyradiculoneuropathy; AMSAN, acute motor and sensory axonal neuropathy; *, verified by nerve conduction studies; NM, not mentioned; D, died; S, survived; I, ineffective; IVIG, intravenous immunoglobulins; GC, glucocorticoid; MV, mechanical ventilator; MRI, magnetic resonance imaging; CT, computed tomography; Abnormal, abnormal enhancement of the cauda equina roots; RD, respiratory distress; ACD, albumincytological dissociation; CIDP, chronic inflammatory demyelinating polyradiculoneuropathy; BBE, Bickerstaff brainstem encephalitis.

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
