# Peer review of "The Peculiarity of Infection and Immunity Correlated with Guillain-Barré Syndrome in the HIV-Infected Population"

_jcm, 2023, doi:10.3390/jcm12030907_

Round 1

Reviewer 1 Report

Thank you for this work.

However, I have a few comments

1)Method: Please specify the date of completion of the data collection. Could you also specify how you did the statistical analyses.

2)For the flowchart, I am not sure I understand: you have 1113+597 items at the beginning and if you exclude the duplicated occurrences, there are 51 items left?

3)In my opinion, the paragraph "The particularity of infectious agents and pathogenesis" is not appropriate for this article: you chose to report cases of HIV infection with GBS. Here it is not a case, unless you specify in each frame, the presence or not of the infectious agent (and the percentage) in the nerve tissue.

4)In the paragraph "The particularity of the immunological spectrum and clinical relevance": specify the number/statistics.

For example: "Among seropositive cases, C. jejuni infection is a rare cause of GBS": specify the percentage of patients with co-infection.

5)Table 1: the legend of the table is incomplete (AGA for antiganglioside antibody?).

6)Table 2: you need to re-specify all abbreviations.

7)When you mention: "The presentation and prognosis of patients with and without HIV infection were identical. In the paragraph: "The particularity of the HIV infection was that the patients with and without HIV infection had the same presentation and prognosis", we would like to have a table to compare the two populations with the statistics.

8)In the paragraph: "The particularity of the presentation and prognosis": specify the number/statistical results.

9)When you write "plasmapheresis, intravenous immunoglobulin (IVIG), and plasma exchange have been applied to treat GBS in patients with HIV, and efficacy has been classified as 1) ineffective, 2) mild benefit, and 3) good response", could you clarify again how you define mild/ineffective/good response as well as the percentage of cases for each class. 

10)You state in the discussion that "the Erasmus Guillain-Barré syndrome respiratory failure score to identify high risk of mechanical ventilation and the modified Erasmus Guillain-Barré syndrome score to predict inability to walk unaided at 6 months, were not applied to HIV positive individuals", however you never mention in the results the number of patients who went to intensive care and how many were intubated. Please add this information to your results or do not conclude on this sentence.

Author Response

Reviewer1

  • Method: Please specify the date of completion of the data collection. Could you also specify how you did the statistical analyses.

Response: Thank you for your kind suggestions.

  • The date of completion of the data collection was November 1, 2022.
  • We used the following terms: (“Guillain-Barré syndrome” or “GBS”, “Inflammatory Demyelinating Polyneuropathy” or “paralysis”, “Acute motor axonal neuropathy”, “Miller-Fisher syndrome” or “ataxia and nystagmus”) and (“Human Immunodeficiency Virus” or “HIV” or “AIDS” or “immune reconstitution”) and searched articles on PubMed and Web of Science databases after 2000. We searched for 597 articles in PubMed and 1113 in Web of Science database. Firstly, we excluded the obvious irrelevant articles in PubMed. Secondly, we excluded the obvious irrelevant articles in Web of Science. Thirdly, we carefully double rechecked the repeated articles in these two databases, and in the end, there were 51 items left. The articles should have case report with definitive diagnosis of GBS. Almost all the recruited cases were verified by nerve conduction studies, which guarantee the more accurate GBS diagnosis. We removed cases with incomplete information and indeterminate diagnostic for GBS. The cases with confirmed chronic inflammatory demyelinating polyradiculoneuropathy (CIDP) were also removed.
  • For the flowchart, I am not sure I understand: you have 1113+597 items at the beginning and if you exclude the duplicated occurrences, there are 51 items left?

Response: Thank you for your kind suggestions.

  • We have 1113+597 items at the beginning, then excluded the obvious irrelevant articles, and carefully double rechecked the repeated articles in these two databases, and in the end, there were 51 items left. For a definitive diagnosis, cases must meet the diagnostic criteria for GBS, such as Brighton criteria laid out by the Brighton Collaboration or the criteria of National Institute of Neurological Disorder and stroke (NINDS)11,12.
  • In my opinion, the paragraph "The particularity of infectious agents and pathogenesis" is not appropriate for this article: you chose to report cases of HIV infection with GBS. Here it is not a case, unless you specify in each frame, the presence or not of the infectious agent (and the percentage) in the nerve tissue.

Response: Thank you for your kind suggestions.

  • Our purpose is to address the peculiarity of infection and immunity among HIV-positive cases with GBS.
  • First, HIV infection is recognized as a infectious agent for all HIV-positive cases and positive plasma HIV RNA at the time of GBS presentation in most cases suggests a para infectious disease mechanism. In Table 1 and Table 2, the lines of HIV status and HIV RNA plasma/ HIV RNA CSF showed the correlated data.
  • Second, CMV and VZV infections are common opportunistic infectious agent for advanced Immunodeficiency cases. CMV infection-associated GBS, which may be verified by positive CMV-DNA in the CSF in one case, as detected by a polymerase chain reaction assay16.A recent herpes zoster attack (reactivation), especially within 1-42 days before GBS onset52, has also been reported in one HIV- positive case8.
  • In the paragraph "The particularity of the immunological spectrum and clinical relevance": specify the number/statistics. For example: "Among seropositive cases, C. jejuni infection is a rare cause of GBS": specify the percentage of patients with co-infection.

Response: Thank you for your kind suggestions. We have specified the numbers in main text according to the information from Table 1 and Table 2, and the following was the description in details:

  • Among the HIV-positive cases, C. jejuni infection is an uncommon cause of GBS, which was only verified in one case25. Other three cases had Escherichia coli, dysentery, enteritis.
  • The common manifestations of ARS are febrility, slight gastrointestinal or respiratory symptoms, and maculopapular eruptions, usually reported 4-30 days before the GBS onset in five cases7,26,27,29,32.
  • The seven cases with IRIS-GBS commonly presents at 1-2 months post- ART19,25,30,39,43,48,50.
  • GBS can also occur when ART is discontinued in four cases17,36,37,46, and GBS episodes coinciding with rebound HIV often present within three months of ART interruption.
  • Another possible mechanism of GBS activation is CSF HIV escape in one case36.
  • Table 1: the legend of the table is incomplete (AGA for antiganglioside antibody?).

Response: Thank you for your kind suggestions.

  • Yes, AGA is for antiganglioside antibody. We have rechecked and improved the legend of the tables.
  • Table 2: you need to re-specify all abbreviations.

Response: Thank you for your kind suggestions. We have already re-specified all abbreviations.

  • When you mention: "The presentation and prognosis of patients with and without HIV infection were identical. In the paragraph: "The particularity of the HIV infection was that the patients with and without HIV infection had the same presentation and prognosis", we would like to have a table to compare the two populations with the statistics.

Response: Thank you for your kind suggestions.

  • We would like to correct the sentence “The presentation and prognosis of patients with and without HIV infection were identical. GBS usually presents as a monophasic course with symptoms peaking within four weeks of onset, followed by a recovery course that persists for months or years” into “GBS usually presents as a monophasic course with symptoms peaking within four weeks of onset, followed by a recovery course that persists for months or years, which were identical between patients with and without HIV infection.”
  • We summarized a table for comparison between HIV-positive GBS cases and general population with GBS.

Characters

HIV-positive

(n=41)

HIV-negative

(cited from References 1 and 18)

Male

31(75.6%)

slightly more frequent in males than in females18

Age > 60

0(0.0%)

 increases with age, especially in

elderly people aged 80 years and over18

Mechanical ventilation

7(17.1%)

20%-30%18

Death

4(9.8%)

3%-7%18

CSF cell count < 50 cells/µL and CSF protein raised

CSF cell count (5-50 cells/µL)

33(80.5%)

Most patients1

15%18

  • In the paragraph: "The particularity of the presentation and prognosis": specify the number/statistical results.

Response: Thank you for your kind suggestions.

  • Among the previously reported HIV-positive GBS cases, most (31cases) involved men and nobody was beyond 60 years.
  • Two patients had a history of GBS19,31.
  • Most HIV-positive patients with GBS (36 cases)
  • Albumin cytologic dissociation (ACD) (33 cases) is a CSF feature typical of GBS where CSF protein levels are elevated due to myelin sheath or axon injury- resulting in increased protein synthesis and release into the CSF, while CSF total white cell count < 50 cells/µL. However, ACD is absent in four HIV-positive patients (CSF total white cell count ≥ 50 cells/µL)32,36,40,41.
  • Typical MRI manifestations are observed in two HIV-positive GBS cases31,47.
  • AGA test was only applied to eleven HIV-positive patients including seven cases with positive results8,10,36.37,41,43,48.
  • When you write "plasmapheresis, intravenous immunoglobulin (IVIG), and plasma exchange have been applied to treat GBS in patients with HIV, and efficacy has been classified as 1) ineffective, 2) mild benefit, and 3) good response", could you clarify again how you define mild/ineffective/good response as well as the percentage of cases for each class. 

Response: Thank you for your kind suggestions.

  • We did not classify the efficacy as 1) ineffective, 2) mild benefit, and 3) good response" for the plasmapheresis, intravenous immunoglobulin (IVIG), and plasma exchange in cases described in Table 1 and Table 2, because the heterogeneity of treatment regimens, dosages and courses in different cases were obvious and some had treatment-related fluctuations (TRFs).If intravenous immunoglobulin (IVIG) or plasma exchange should be started as soon as possible, before irreversible nerve damage has taken place. We would like to correct this sentence” efficacy has been categorized as 1) ineffective, 2) mild benefit, and 3) good response” into “, which should be started as soon as possible.”.
  • In Table 1 and Table 2, the last line is final outcome(prognosis),which was classified by D/S/I(D, died; S, survived; I, ineffective).
  • Second, among cases of IRIS-related GBS, which is associated with cell-mediated immune mechanisms53, steroid initiation was applied to two cases19,30.
  • We added the sentence “Finally, during the progressive phase, patients with the possibility of developing respiratory failure and requiring ventilation should be timely transferred to an intensive care unit”.
  • You state in the discussion that "the Erasmus Guillain-Barré syndrome respiratory failure score to identify high risk of mechanical ventilation and the modified Erasmus Guillain-Barré syndrome score to predict inability to walk unaided at 6 months, were not applied to HIV positive individuals", however you never mention in the results the number of patients who went to intensive care and how many were intubated. Please add this information to your results or do not conclude on this sentence.

Response: Thank you for your kind suggestions.

  • During the acute phase (symptoms usually peak within 4 weeks and progression can last up to 6 weeks after onset), it is important using the Erasmus Guillain-Barré syndrome respiratory failure score at hospital admission to identify high risk of mechanical ventilation. During the stable phase or even the recovery period (can last months or years), it is important using the modified Erasmus Guillain-Barré syndrome score at hospital admission and at day 7 of admission to predict inability to walk unaided at 6 months. However, only several cases were assessed by these scoring system. Therefore, the performance of GBS prognostic models should be further validated among HIV-positive individuals.
  • Among a total of 41 cases, 7 cases were intubated, which has been described as MV, mechanical ventilator in the line of Treatment. Autonomic disturbance-related cardiac arrest and respiratory failure requiring mechanical ventilation (seven cases) are the most common cause of death (four cases)7,17,23,24,27,38,39,44.

Reviewer2

Comments and Suggestions for Authors

Thanks to the authors for the manuscript. They describe the features of GBS in HIV-infected population. The review is well written and understandable by any physician. It is a good article to read and consult.

 The authors describe Guillain-Barré syndrome in HIV population. The topic is interesting, because the review provides a good summary of an increasingly common pathology in a specific population, and because there are few few published cases. The authors do a review the literature published since 2000, and summarized the clinical manifestations and diagnostic examinations, with some evident conclusions that synthesize well what has been studied.In my opinion the different article sections are correct in length and content. The method and flowchart are adequate and allows us to perform the same search. The references are appropriate and up to date and the tables help to understand the study.

Response: Thank you very much.

Reviewer 2 Report

Thanks to the authors for the manscript. They describe the features of GBS in HIV-infected population. The review is well written and understandable by any physician. It is a good article to read and consult.

The authors describe Guillain-Barré syndrome in HIV population. The topic is interesting, because the review provides a good summary of an increasingly common pathology in a specific population, and because there are few few published cases. The authors do a review the literature published since 2000, and summarized the clinical manifestations and diagnostic examinations, with some evident conclusions that synthesize well what has been studied.

In my opinion the different article sections are correct in length and content. The method and flowchart are adequate and allows us to perform the same search. The references are appropriate and up to date and the tables help to understand the study.

Author Response

Reviewer2

Comments and Suggestions for Authors

Thanks to the authors for the manuscript. They describe the features of GBS in HIV-infected population. The review is well written and understandable by any physician. It is a good article to read and consult.

 The authors describe Guillain-Barré syndrome in HIV population. The topic is interesting, because the review provides a good summary of an increasingly common pathology in a specific population, and because there are few published cases. The authors do a review the literature published since 2000, and summarized the clinical manifestations and diagnostic examinations, with some evident conclusions that synthesize well what has been studied. In my opinion the different article sections are correct in length and content. The method and flowchart are adequate and allows us to perform the same search. The references are appropriate and up to date and the tables help to understand the study.

Response: Thank you very much.

Round 2

Reviewer 1 Report

Thanks you for all these corrections. 

Just one last thing : Because of several missing figures, the table comparing HIV and non-HIV patients is not very interesting. 

Thank you for this paper.